Factors influencing catheter-related infections in peritoneal dialysis patients: a meta-analysis

Sun Juan sunjuan@xxmu.edu.cn 1
Zhao Miaomiao 1
Zhang Yifang 1
Zhang Wenyu 1
Zhang Huimin 1
Bao Qingzhu 2
1 School of Nursing, Xinxiang Medical University , Xinxiang , Henan , China
2 Xinxiang First People’s Hospital , Xinxiang , Henan , China
Capusa Cristina
Electronic publication date: 2025 Sep 29
Publication date: 2025
Volume: 13
Electronic Location ID: e20063
Received 2025 Feb 27; Accepted 2025 Aug 20
Copyright: ©2025 Sun et al.
Copyright year: 2025
Copyright holder: Sun et al.
License: This is an open access article distributed under the terms of the Creative Commons Attribution License, which permits unrestricted use, distribution, reproduction and adaptation in any medium and for any purpose provided that it is properly attributed. For attribution, the original author(s), title, publication source (PeerJ) and either DOI or URL of the article must be cited.
License URL: https://creativecommons.org/licenses/by/4.0/

Keywords: Chronic kidney disease, Peritoneal dialysis, Catheter-related infections, Evidence-based nursing, Meta-analysis

Funding: Science and Technology Tackling Project of Henan Province in 2024 242102310319 Humanities and Social Sciences Research Project of the Ministry of Education of Henan Province 2024-ZDJH-485 Henan Province Higher Education Key Scientific Research Project 23B320002 Quality Courses for Postgraduate Students in Henan Province, 2024 YJS2024KC24 This work was supported by the Science and Technology Tackling Project of Henan Province in 2024 (242102310319); Humanities and Social Sciences Research Project of the Ministry of Education of Henan Province (2024-ZDJH-485); Henan Province Higher Education Key Scientific Research Project (23B320002); Quality Courses for Postgraduate Students in Henan Province, 2024 (YJS2024KC24). The funders had no role in study design, data collection and analysis, decision to publish, or preparation of the manuscript.

==============================
Objective

Systematic analysis of factors influencing catheter-related infections in patients undergoing peritoneal dialysis.

Methods

A computerized search of Chinese and English databases was conducted to identify the factors affecting catheter-related infections in patients undergoing peritoneal dialysis. Studies with a search period from inception to July 31, 2024 were retrieved. Two researchers independently screened the literature using inclusion and exclusion criteria, and data extraction and quality assessments were performed. Meta-analysis was performed using RevMan5.4 and Stata software.

Results

In total, 13 studies were included and meta-analysis revealed that comorbid diabetes mellitus, insecure catheter fixation, dialysis duration >2 years, body mass index >20 kg/m2, mechanical strain, lack of proper care, irregular caregivers, and history of catheter pulling (P < 0.05) were the influencing factors contributing to the incidence of catheter-related infections in patients undergoing peritoneal dialysis.

Conclusions

Clinicians should pay close attention to these influencing factors and provide targeted preventive measures to prevent catheter-related infections.

Introduction

Chronic kidney disease (CKD) is one of the most serious diseases affecting humans health worldwide (Guo et al., 2023) with a global prevalence of approximately 11–13% (Hill et al., 2016). CKD is characterized by high morbidity and disability rates; high medical costs; and low prevention, treatment, and awareness rates, resulting in many patients with CKD progressing to stage 5 CKD and end-stage kidney disease by the time they are diagnosed with CKD (Lin et al., 2020a; Lin et al., 2020b). Peritoneal dialysis (PD) is recognized worldwide and widely used kidney replacement therapies for patients with end-stage kidney disease (Rivara & Mehrotra, 2014; Shrestha et al., 2018; Yue et al., 2019; Guo et al., 2019; Ye et al., 2019; Htay et al., 2020; Wang et al., 2021).

However, most patients undergoing PD are treated and cared at home without the guidance and supervision of healthcare professionals. Dialysis itself stimulates the body to increase catabolism, which is prone to nutritional risks, peritonitis, and catheter-related infections—such as exit-site infection (ESI) and tunnel infection (TI) that often co-exist (Pan, Li & Miao, 2021). The most common causes of catheter infections are frequent friction at the exit site, pressure, and failure to clean the exit site; infections are influenced by various factors and will further progress to peritonitis if not treated promptly (Van Diepen & Jassal, 2013). PD-associated infections were identified as the most important clinical outcomes in PD, as stated in the Standardized Outcomes in Nephrology–PD (SONG-PD), which can lead to hospitalization, transfer to hemodialysis, and death (Pan, Li & Miao, 2021). Despite their importance, the rate and outcome of PD-associated infections vary worldwide and have been largely attributed to facility-level factors (i.e., practices and organizational structures within healthcare facilities) (Van Diepen & Jassal, 2013). Although the incidence of these infections is low, they can have serious adverse effects on patient prognosis. Therefore, it is particularly important to understand the factors that contribute to the occurrence of infections in patients and explore the reasons for their occurrence. Relevant domestic and international studies have been conducted regarding the risk factors of catheter-related infections in patients undergoing PD. Some studies have reported that infections are associated with multiple factors (e.g., poor catheter care and unsecured catheters); however, the results of these studies are contradictory and have not yet been clearly established. Therefore, this study conducted a meta-analysis to explore the risk factors of catheter-related infections in patients undergoing PD, aiming to identify the independent risk factors and provide theoretical guidance for future targeted intervention measures.

Review methods

Inclusion and exclusion criteria

Inclusion criteria

(1) The study population was patients undergoing PD,

(2) Exposure factors: influential factors associated with catheter-related infections in patients undergoing PD,

(3) Types of studies: case-control and cohort studies.

Exclusion criteria

(1) Repeat publication,

(2) Literature in which the full text is not available or in which the data is incorrect,

(3) Reviews, animal experiments, conference abstracts, and case reports,

(4) Meta-randomized controlled trials (RCTs).

Search strategy

The review was preregistered and was enrolled in PROSPERO (study ID: CRD42023478459). The reporting complied with the PRISMA (Preferred Reporting Items for Systematic Reviews and Meta-Analyses) statement and AMSTAR (A Measurement Tool to Assess Systematic Reviews) guidelines. The electronic databases searched included CNKI, VIP, WanFang, PubMed, Web of Science, Cochrane Library, CINAHL, Embase, and MEDLINE. Platform and topic-sensitive search strategies were created using medical subject headings (MeSH) and common terms (File S1). To ensure the feasibility and manageability of our meta-analysis, we limited our search to studies published in English or Chinese language. This decision was based on the primary language proficiency of our target audience and the substantial body of evidence available in these languages. We acknowledge that this approach may have excluded relevant studies published in other languages; however, we believe that the included studies provide a comprehensive and representative overview of the current evidence on the risk factors for catheter-related infections in patients undergoing PD. In this meta-analysis, the search strategy was executed by two authors (Juan Sun and Miaomiao Zhao) independently, who screened the study titles and abstracts retrieved from databases such as PubMed, Embase, and the Cochrane Library based on predefined inclusion and exclusion criteria. Any disagreements regarding the study eligibility were resolved through discussion by the two authors. If a consensus could not be reached, a third author, Qingzhu Bao, served as an arbitrator to make the final decision. This process ensured rigor and objectivity of study selection, and all disagreements and resolutions were documented to maintain transparency and reproducibility in accordance with the meta-analysis guidelines.

Study selection and data extraction

Two researchers independently screened the literature, extracted data, and crosschecked them based on the inclusion and exclusion criteria. We focused on case-control and cohort studies and excluded RCTs. This decision was based on the nature of our research question, which aimed to identify the risk factors for catheter-related infections in patients undergoing PD. RCTs are typically designed to evaluate interventions rather than explore risk factors, whereas case-control and cohort studies are more suitable for examining the associations between potential risk factors and outcomes. Additionally, the limited availability of RCTs addressing this specific topic and the broader clinical context of patients undergoing PD further justified our choice of study design. Data extraction included: first author, publication date, country, type of study, sample size, influencing factors, and information related to quality assessment of the literature; data were retrieved from the included papers. To ensure transparency and reproducibility, the data extracted in this study are publicly available as File S3.

Quality assessment

The Newcastle-Ottawa Quality Assessment Scale (NOS) was used to assess the quality of case-control and cohort studies (Stang, 2010). The scale comprises three categories: study population selection, between-group comparability, and exposure or outcome evaluation, with a total of 9 points distributed across eight entries. The score of 0–4 indicates low quality, 5–6 indicates moderate quality, and 7–9 indicates high quality (Ga, 2000). This scale assesses study quality by evaluating eight elements related to participant selection, comparability of research groups, and determination of outcomes or exposure. Two independent researchers conducted a critical assessment of the methodological quality and risk of bias of the included studies. In cases of disagreement, consensus was reached after discussion with a third party.

Statistical methods

Stata version 17 and Revman 5.4 were utilized for all analyses. The effect sizes are presented as odds ratios (ORs) with 95% confidence intervals (CIs) and a significance level of P < 0.05, which was considered statistically significant. For assessing heterogeneity, P > 0.10 and I2 ≤ 50% suggested low heterogeneity, prompting the selection of the fixed-effects model for meta-analysis. Conversely, P < 0.10 and I2 ≥ 50% indicated high heterogeneity, leading to the adoption of random-effects model for analysis. Sensitivity analyses were performed by changing the data analysis model as well as single-article exclusion, and Egger’s test was used to detect publication bias for studies with more than two articles. The difference was considered statistically significant at P < 0.05, to determine the reliability and stability of the results.

Results

Literature search process and results

The initial search yielded 1,289 articles of relevant literature, and tracing references yielded two additional pieces, resulting in a total of 1,291 articles. Following a systematic screening process, 13 studies (Yue et al., 2019; Xu et al., 2019; Lin et al., 2020a; Lin et al., 2020b; Liu et al., 2021; Cao et al., 2021; Yang, Wu & Wang, 2021; Ding et al., 2021; Wang, 2021; Lu et al., 2022; Huang et al., 2022; Wong et al., 2022; Yang et al., 2023) were ultimately included, as depicted in Fig. 1. Three studies were excluded because the data could not be extracted (Menegueti et al., 2017; Yap et al., 2018; Cheng et al., 2019), and one study was excluded because the results revealed no significance (Zhang et al., 2018). All exclusion rationales were documented prospectively. The full details of the excluded studies are provided in the File S4.

Figure 1 PRISMA flow diagram.

The 13 studies that were included comprised 10 case-control studies and three cohort studies involving 4,215 study participants. The methodological quality of the included 13 papers was evaluated using the NOS scale (Ga, 2000), and all the included literature scored ≥5. The basic characteristics of the included studies and results of the methodological quality evaluation are shown in Table 1.

Table 1 Basic characteristics of included studies and results of methodological quality assessment.

Inclusion of literature	Published	States	Type of study	Sample size (cases)	Risk factor	Quality assessment (points)	
Yue et al. (2019)	2019	China	Case-control	108	➃	5	
Xu et al. (2019)	2019	China	Case-control	236	➀➁➃➄	6	
Lin et al. (2020a)	2020	China	Cohort	1,133	➂➅➈	7	
Ding et al. (2021)	2021	China	Case-control	247	➀➈➉	7	
Lin et al. (2020b)	2020	China	Cohort	1,204	➂➇➉	6	
Cao et al. (2021)	2021	China	Case-control	70	➀	6	
Wang et al. (2021)	2021	China	Case-control	40	➀➁➃➄➆	5	
Yang, Wu & Wang (2021)	2021	China	Case-control	154	➂	7	
Liu et al. (2021)	2021	China	Case-control	208	➈	7	
Wong et al. (2022)	2022	Malaysia	Cohort	201	➃	5	
Lu et al. (2022)	2022	China	Case-control	213	➇	6	
Huang et al. (2022)	2022	China	Case-control	273	➀	7	
Yang et al. (2023)	2023	China	Case-control	128	➀➁➂➃➄➅➆	6	
Notes.

➀Diabetes ➁Hypertension ➂Catheter fixation ➃Dialysis time ➄Body mass index (BMI) ➅Mechanical stress ➆Unstandardized guarding ➇Caregiver immobilization ➈History of catheter pulling ➉Care process adherence Albumin level.

Meta-analysis results

In this study, 13 articles were examined to identify the factors that influenced catheter-related infections in patients undergoing PD. We combined articles with two or more influencing factors in common. The meta-analysis results revealed that diabetes mellitus (DM), insecure catheter placement, dialysis duration >2 years, body mass index (BMI) >20 kg/m2, mechanical strain, lack of proper care, irregular caregivers, and a history of catheter pulling were significant factors influencing catheter-related infections in patients undergoing PD (P < 0.05). The meta-analysis results are detailed in Table 2. The detailed forest and Egger’s test plots for each analysis are shown in Figs. 2–14.

Figure 2 Forest plots for combined diabetes.

Sources: Cao et al., 2021; Ding et al., 2021; Huang et al., 2022; Wang et al., 2021; Xu et al., 2019; Yang et al., 2023.

Figure 3 Forest plots for combined hypertension.

Sources: Wang et al., 2021; Xu et al., 2019; Yang et al., 2023.

Figure 4 Forest plots for insecure catheter placement.

Sources: Lin et al., 2020a; Lin et al., 2020b; Yang, Wu & Wang, 2021; Yang et al., 2023.

Figure 5 Forest plots for dialysis duration.

Sources: Wang et al., 2021; Xu et al., 2019; Yang et al., 2023.

Figure 6 Forest plots for BMI > 20 kg/m2.

Sources: Wang et al., 2021; Xu et al., 2019; Yang et al., 2023.

Figure 7 Forest plots for mechanical pressure.

Sources: Lin et al., 2020a; Lin et al., 2020b; Yang et al., 2023.

Figure 8 Forest plots for lack of proper care.

Sources: Wang et al., 2021; Yang et al., 2023.

Figure 9 Forest plots for irregular caregivers.

Sources: Lin et al., 2020a; Lin et al., 2020b; Lu et al., 2022.

Figure 10 Forest plots for history of catheter pulling.

Sources: Ding et al., 2021; Lin et al., 2020a; Lin et al., 2020b; Liu et al., 2021.

Figure 11 Egger’s publication bias plot for combined diabetes.

Figure 12 Egger’s publication bias plot for insecure catheter placement.

Figure 13 Egger’s publication bias plot for history of catheter pulling.

Figure 14 Egger’s publication bias plot for dialysis duration > 2 years.

Combined diabetes

Six studies (Xu et al., 2019; Cao et al., 2021; Ding et al., 2021; Wang, 2021; Huang et al., 2022) reported a relationship between patients undergoing PD with comorbid diabetes and the occurrence of catheter-related infections. All six studies were performed in case-control groups and involved 681 patients with comorbid diabetes and 1,216 patients without comorbid diabetes. The pooled results showed a large heterogeneity among the studies (P < 0.001, I2 = 85%), and a meta-analysis was performed using random effects. The results showed that patients with comorbid DM had a 2.52 times higher risk of catheter-related infections than those without comorbid DM (OR = 2.52; 95% CI [1.44–4.42]; P = 0.001; Fig. 2). The Egger publication bias test for the included studies showed no publication bias (P > 0.05; Fig. 11).

Combined hypertension

Three studies (Xu et al., 2019; Wang, 2021; Yang et al., 2023) reported the effect of combined hypertension on the occurrence of catheter-related infections in patients undergoing PD. Heterogeneity among the studies was high (P = 0.005, I2 = 81%), and meta-analysis was performed using a random-effects model. The combined results showed that comorbid hypertension was not an influential factor for catheter-related infection in patients undergoing PD (OR = 2.25; 95% CI [0.97–5.23]; P = 0.06; Fig. 3). The source of heterogeneity was analyzed using the literature-by-exclusion method, and the sensitivity analysis results showed that heterogeneity declined to 0% after excluding the study by Xu et al. (2019) and P < 0.001, which was an unstable result (Table 3).

Table 2 Results of meta-analysis of factors influencing catheter-related infections in peritoneal dialysis patients.

Risk factors	Inclusion of studies	Heterogeneity test	Effect model	Meta-analysis results	
		I 2	P		OR	95% CI	P	
Combined diabetes	6	85%	<0.001	Random	2.52	[1.44–4.42]	0.001	
Combined hypertension	3	81%	0.005	Random	2.25	[0.97–5.23]	0.06	
Insecure catheter placement	4	61%	0.05	Random	2.52	[1.42–4.48]	0.002	
Dialysis duration > 2 years	3	71%	0.03	Random	5.68	[2.95–10.93]	<0.001	
BMI > 20 kg/m2	2	0%	0.52	Fixed	5.05	[3.25–7.85]	<0.001	
Mechanical pressure	2	0%	0.06	Fixed	5.20	[2.69–10.05]	<0.001	
lack of proper care	2	0%	0.37	Fixed	5.91	[4.14–8.45]	<0.001	
Irregular caregivers	2	0%	0.76	Fixed	2.65	[1.37–5.15]	0.004	
History of catheter pulling	3	46%	0.15	Fixed	2.55	[1.94–3.36]	<0.001	
Notes.

Random: Random-effects model; Fixed: Fixed-effects model.

Insecure catheter placement

Four studies (Lin et al., 2020a; Lin et al., 2020b; Yang, Wu & Wang, 2021; Yang et al., 2023) reported the effect of poor catheter fixation on the occurrence of catheter-related infections in patients undergoing PD. Heterogeneity among the studies was high (P = 0.05, I2 = 61%), and meta-analysis was performed using a random-effects model. The combined results showed that poor catheter fixation was an influential factor for catheter-related infections in patients undergoing PD (OR = 2.52; 95% CI [1.42–4.48]; P = 0.002; Fig. 4), and results of the Egger’s test showed no publication bias. Sensitivity analysis showed that after excluding the study by Yang et al. (2023), heterogeneity decreased to 0% and P < 0.001, and the results were unstable (Table 3).

Dialysis duration

Five studies (Yue et al., 2019; Xu et al., 2019; Wang, 2021; Wong et al., 2022; Yang et al., 2023) reported the effect of dialysis duration on the occurrence of catheter-related infections in patients undergoing PD. However, in a study by Yue et al. (2019), the duration of dialysis was categorized as ≤2 years, 12–36 months, and >36 months, which did not allow for data merging. Wong et al. (2022) reported a 17% increase in risk for each month of decrease in PD duration, which is a protective factor, and exclusion from data merging occurred because there was only one study. Heterogeneity between studies was large (P = 0.03, I2 = 71%), and meta-analysis was performed using a random-effects model. The combined results showed that the risk of catheter-related infection in patients undergoing PD with >2 years of dialysis was 5.68 times higher than that of those with ≤2 years of dialysis (OR = 5.68; 95% CI [2.95–10.93]; P < 0.001; Fig. 5). The sensitivity analysis showed more stable results.

BMI > 20 kg/m2

Three studies (Xu et al., 2019; Wang, 2021; Yang et al., 2023) reported the effect of BMI on the occurrence of catheter-related infections in patients undergoing PD. Heterogeneity among the studies was large (P = 0.003, I2 = 82%), and a meta-analysis was performed using a random-effects model. The combined results showed that patients undergoing PD with BMI > 20 kg/m2 had 6.81 times higher risk of catheter-related infection than those with BMI ≤ 20 kg/m2 (OR = 6.81; 95% CI [3.51–13.20]; P < 0.001; Fig. 6). The sensitivity analysis results showed a decrease in heterogeneity after the exclusion of the study by Xu et al. (2019) (I2 = 0%, P = 0.52), and the results were unstable (Table 3).

Mechanical pressure

Two studies (Lin et al., 2020a; Lin et al., 2020b; Yang et al., 2023) reported the effect of BMI on the occurrence of catheter-related infections in patients undergoing PD. The heterogeneity among the studies was small (P = 0.68, I2 =0%), and the meta-analysis was performed using a fixed-effects model. The combined results showed that mechanical stress was an influential factor in the development of catheter-related infections in patients undergoing PD (OR = 5.20; 95% CI [2.69–10.05]; P < 0.001; Fig. 7).

Lack of proper care

Two studies (Wang, 2021; Yang et al., 2023) reported the effect of inadequate protection on the occurrence of catheter-related infections in patients undergoing PD. The heterogeneity between the studies was small (P = 0.37, I2 = 0%), and the meta-analysis was performed using a fixed-effects model. The combined results showed that unregulated protection was an influential factor in the occurrence of catheter-related infections in patients undergoing PD (OR = 5.91; 95% CI [4.14–8.45]; P < 0.001; Fig. 8).

Irregular caregivers

Two studies (Lin et al., 2020a; Lin et al., 2020b; Lu et al., 2022) reported the effect of irregular caregiver on the occurrence of catheter-related infections in patients undergoing PD. The heterogeneity among the studies was small (P = 0.76, I2 = 0%), and the meta-analysis was performed using a fixed-effects model. The combined results showed that irregular caregiver was an influential factor in the occurrence of catheter-related infections in patients undergoing PD (OR = 2.65; 95% CI [1.37–5.15]; P = 0.004; Fig. 9).

History of catheter pulling

Three studies (Lin et al., 2020a; Lin et al., 2020b; Liu et al., 2021; Ding et al., 2021) reported the effect of a history of catheter pulling on the occurrence of catheter-related infections in patients undergoing PD. The studies reported low heterogeneity (P = 0.15, I2 = 46%). The combined results showed that a history of catheter pulling was an influential factor in the development of catheter-related infections in patients undergoing PD (OR = 2.55; 95% CI [1.94–3.36]; P < 0.001; Fig. 10).

Publication bias and sensitivity analysis

Changes in the combined effect sizes were observed by excluding any of the included studies, and the results showed that hypertension, insecure catheter placement, and BMI > 20 kg/m2 showed significantly lower heterogeneity after excluding individual studies. History of catheter pulling showed significantly higher heterogeneity after excluding individual studies, and the remaining influencing factors remained unchanged, as shown in Table 3. In addition, we conducted a comparative analysis of the differences before and after the use of the altered data analysis models for these 11 influencing factors, with statistically significant combined effect sizes. The results showed no significant changes in the combined effect sizes of the 11 influencing factors, and the results were relatively stable, as shown in Table 4. We conducted the Egger’s test to assess publication bias for the risk factors examined in more than two studies. Specifically, we performed these tests for the following factors: comorbid diabetes (Fig. 11), insecure catheter placement (Fig. 12), history of catheter pulling (Fig. 13), and dialysis duration >2 years (Fig. 14), as shown in (Table 4). No significant publication bias regarding these risk factors was detected among the included studies (P > 0.05).

Table 3 Results of exclusion analysis of factors affecting catheter-related infections in peritoneal dialysis patients.

Risk factors	Model	Exclusion studies	Before excluding	After excluding	
			Heterogeneity (%)	OR (95% CI)	P	Heterogeneity (%)	OR (95% CI)	P	
Combined hypertension	Random	Xu, YL	81	2.25 [0.97–5.23]	0.06	0	3.47 [1.89–6.40]	<0.001	
Insecure catheter placement	Random	Yang JB	61	2.52 [1.42–4.48]	0.002	0	1.87 [1.35–2.58]	<0.001	
BMI > 20kg/m2	Random	Xu, YL	82	6.81 [3.51–13.20]	<0.001	0	5.05 [3.25–7.85]	<0.001	

Table 4 Sensitivity, Egger’s test analysis of factors affecting catheter-related infections in peritoneal dialysis patients.

Risk factors	Inclusion of studies	Before Sensitivity Analysis	After sensitivity analysis	Stability	Egger’s test	
		Model	OR	95% CI	P	Model	OR	95% CI	P			
Combined diabetes	6	Random	2.52	[1.44–4.42]	0.001	Fixed	2.58	[2.15–3.09]	<0.001	stability	0.744	
Insecure catheter placement	4	Random	2.52	[1.42–4.48]	0.002	Fixed	2.08	[1.52–2.84]	<0.001	stability	0.096	
Dialysis duration > 2 years	3	Random	5.68	[2.95–10.93]	<0.001	Fixed	4.62	[3.40–6.28]	<0.001	stability	0.207	
BMI > 20 kg/m2	2	Fixed	5.05	[3.25–7.85]	<0.001	Random	5.05	[3.25–7.85]	<0.001	stability	–	
Mechanical pressure	2	Fixed	5.20	[2.69–10.05]	<0.001	Random	5.20	[2.69–10.05]	<0.001	stability	–	
lack of proper care	2	Fixed	5.91	[4.14–8.45]	<0.001	Random	5.91	[4.14–8.45]	<0.001	stability	–	
Irregular caregivers	2	Fixed	2.65	[1.37–5.15]	0.004	Random	2.65	[1.37–5.15]	0.004	stability	–	
History of catheter pulling	3	Fixed	2.55	[1.94–3.36]	<0.001	Random	2.59	[1.64–4.08]	0.001	stability	0.772	
Notes.

Random: Random-effects model; Fixed: Fixed-effects model; Egger’s test was performed only for influences that included more than 2 papers.

Discussion

Better methodological quality of included studies

This study used the results of a multifactor logistic regression analysis to reduce the confounding effects. The study systematically evaluated the literature screening and inclusion in strict accordance with evidence-based methods to ensure the credibility of the findings. Most of the included studies were case-control studies, which generally have weaker causal arguments than other study designs. Additionally, only three studies were published in English, and these studies had average sample representativeness. These factors may have affected the quality of the meta-analysis. Of the 13 studies, eight were rated as medium-quality and five as high-quality. Although the quality of the studies requires further improvement, the overall quality was deemed satisfactory, and the results were considered credible. Our study included several influencing factors identified in the available literature; however, the diversity of these factors may be limited by the scope of our study. These findings provide valuable insights into the risk factors of catheter-related infections in patients undergoing PD.

Factors associated with increased infection risk

Comorbid diabetes

DM and CKD pose major challenges to global public health and their associated complications have a significant impact on individuals and healthcare systems worldwide (Chow et al., 2023). Poorly managed DM is a recognized risk factor of cardiovascular events and mortality in patients undergoing dialysis (Iida et al., 2018). Patients with DM develop conditions such as low immunity, poor nutritional status, and poor tolerance, which predispose them to complications such as malnutrition and secretory disorders (Yap et al., 2018; Cheng et al., 2019; Cao et al., 2021), increasing the risk of infections in patients undergoing PD. Recently, it has been suggested that PD treatment may increase the risk of developing new-onset DM (NODM) in patients with end-stage renal disease, especially in those with preexisting prediabetes. These findings underline the importance of a personalized approach to treatment; therefore, for patients undergoing PD, treating the underlying disease and controlling blood glucose levels are crucial in preventing catheter-related infections. This finding also suggests that prediabetes should be considered when choosing dialysis modalities for the patients by the nephrologists (Chen et al., 2024).

One limitation of our study is that we did not distinguish between Type 1 and Type 2 DM. This can be attributed to the limitations in the data available in the included studies, many of which did not differentiate between the types of diabetes. Future research should aim to investigate the distinguishing effects of Type 1 and Type 2 diabetes on the risk of catheter-related infections in patients undergoing PD, as this could provide valuable insights for targeted prevention and management strategies.

Dialysis duration > 2 years

As an invasive operation, PD catheterization will destroy the protective barrier of the patient’s skin, leading to a decline in the body’s resistance; prolonged retention of the catheter will lead to easy bacterial invasion of the body along the catheter, followed by bacterial proliferation and infection (Yap et al., 2018; Guo et al., 2019). With the prolongation of the retention time, its surface forms a layer of loose fibrin sheath, which provides a favorable environment for the propagation, migration, and adhesion of bacteria at the puncture site and limits the action of host phagocytes and antibacterial drugs, which can lead to infection (Zhang et al., 2018). In addition, the dialysis time is too long and patients are prone to irritability and boredom owing to repeated operations, excessive economic burden, aseptic awareness of indifference, and decreased compliance with routine nursing operations, which in turn cause infections. Therefore, for patients with a dialysis duration >2 years, retraining of disease-related knowledge is crucial (Chow et al., 2023). However, Yue et al. (2019) categorized dialysis duration into three stages, namely <12 months, 12–36 months and >36 months, to explore the risk of infection at different time stages, and revealed that catheter-related infections were least likely to occur at dialysis duration of 12–36 months, and the chance of infections increased at <12 and >36 months. Therefore, retraining disease-related knowledge is essential for patients undergoing dialysis for >2 years (Chow et al., 2023). Self-catheterized exit site care, nurse exit site training, and retraining are important in the prevention of PD-related infections (Bernardini, Price & Figueiredo, 2006; Iida et al., 2018). Most of the post-discharge continuing education is accomplished through follow-up visits, of which home visits play a key role. A relatively sound home-visit model exists in foreign countries; however, the home-visit model in China is still in its infancy (Lu & Xing, 2017). In the future, home visits and nursing knowledge retraining for patients undergoing long-term PD should be strengthened.

BMI > 20 kg/m2

The results of a large cohort study in the United States indicated that obese patients undergoing PD had a higher risk of complications than nonobese patients (Obi et al., 2018), consistent with our study results. However, Xu et al. reported that BMI ≤20 kg/m2 was a risk factor for the development of catheter-related infections (Xu et al., 2019), contrary to the results of our study. Another study indicated that weight gain was common in the first 2 years of PD, but it did not seem to have any significant effect on subsequent outcomes. Contrastingly, >5% weight loss was significantly associated with poor patient survival (Than et al., 2023). Currently, there are only few studies on the effect of BMI on PD catheter-related infections, and only three studies were included in this review; therefore, whether BMI > 20 kg/m2 isan influential factor for catheter-related infections in patients undergoing PD is still controversial and needs to be further explored.

Lack of proper care, insecure catheter placement, and mechanical pressure

PD is a long-term treatment, and most patients are treated and cared for at home, lacking the guidance and supervision of healthcare personnel. This leads to a reduced sense of protection and a weak concept of asepsis, coupled with the fact that dialysis itself stimulates an increase in the body’s catabolism and metabolism, which makes it prone to nutritional risks, peritonitis, and catheter-related infections (You et al., 2024). Therefore, in the future, healthcare staff should increase the frequency of training and follow-up visits to detect protective irregularities in a timely manner and reduce the occurrence of catheter-related infections. The results of this study showed that poor catheter fixation and mechanical pressure are among the causes of catheter-related infections in patients undergoing PD, which may be attributed to the fact that the abdominal dialysis catheter is placed in the patient’s tunnel for a long period and is susceptible to friction if it is poorly fixed. If the catheter is over-fixed or mechanically compressed by the belt, it will lead to delayed healing at the exit, which will lead to an increase in the risk of infection (Lai et al., 2018). However, such events are preventable, as they are largely dependent on controllable factors such as appropriate dialysis catheter care, which can be addressed through effective health education (Béchade et al., 2017). Ma et al. (2020) reported that inadequate training of PD catheters may be responsible for catheter-related mechanical problems (e.g., catheter dysfunction, bleeding, and hematomas) and infectious complications (e.g., peritonitis), which can reduce the effectiveness of dialysis treatment and negatively impact patient mortality. Therefore, effective patient education is essential to ensure that patients possess the necessary knowledge and skills required for peritoneal catheter care (Devoe et al., 2016). Nursing staff should train patients in catheter care and supervise the proper placement of PD tubing, regularly observe the skin at the catheter exit site, and avoid catheter pressure and skin friction to avoid the incidence of catheter-related adverse events (Chow et al., 2023).

Irregular caregivers

PD can be self-administered at home and, to some extent, allows patients to maintain a good quality of life (Hiramatsu et al., 2020). However, PD increases the burden on the patients owing to factors such as declining health status and the disease itself, and often requires dependence on caregivers and nursing staff (Jiang & Zheng, 2022). The results of this study showed that irregular caregivers are a risk factor for catheter-related infection in patients undergoing PD. The reason for this may be that most patients undergoing PD are taken care of by a person who is most familiar with the disease, asepsis awareness, and care procedures. It is important to note that variations in the degree of expertise among nursing personnel in relation to aseptic procedures, catheter fixation, and the management of the exit site may potentially lead to inadequate procedures or the failure to perform essential steps if there is a change in nursing staff midway through the process. Furthermore, even if nursing staff have received training, if they have not performed the procedures for a considerable amount of time or lack ongoing retraining, they may experience a decline in knowledge and become unacquainted with the procedures, leading to a deterioration in the quality of care.

In addition, family caregivers have been found to help patients with the activities of daily living. Unfortunately, patients’ quality of life declines over time, and various methods have been explored to address this issue (Hovadick et al., 2021). A single-center study found that home training for patients on continuous PD and four home visits for patients with low autonomy showed that this protocol increased the level of autonomy and patients’ own potential (Milan Manani et al., 2024). Although this study integrated existing evidence and found that inconsistent nursing staff may be one of the factors influencing catheter exit site infections in patients undergoing PD at home, the number of original studies on this factor is limited. This study only included two studies, with insufficient sample size and weak evidence strength. Therefore, high-quality original studies involving large-samples are needed in the future to further validate this association.

History of catheter pulling

Bleeding from pulling catheters is mostly due to insecure fixation, excessive activity, or turning over at night without paying attention to the catheter, and long-term bleeding from pulling is prone to chronic infection of the local skin. Therefore, more attention should be paid during catheter fixation, avoid pulling of catheters, and communicate promptly with the medical staff for the effective treatment of patients with bleeding to prevent further deterioration and infection (Yu et al., 2008).

Relevance for clinical practice

Implications for clinical practice

In clinical practice, the rapid development of healthcare services and the continuous improvement of medical standards have led to a significant decrease in the incidence of catheter-related infections. However, the problem of catheter-related infections due to prolonged tube placement and low levels of self-management in patients undergoing PD is still prominent. The results of this study showed that a combination of DM, insecure catheter placement, dialysis duration > 2 years, BMI > 20 kg/m2, mechanical strain, lack of proper care, irregular caregivers, and a history of catheter pulling were risk factors for the development of catheter-related infections in patients undergoing PD. This study identified risk factors for catheter-related infections in PD, which are important for healthcare professionals to consider in clinical practice. Healthcare professionals can effectively identify patients at a high risk of developing catheter-related infections based on the outcome elements of this study and implement personalized and targeted interventions to reduce the risk of infection.

In this review, we have summarized the influential factors, including both controllable and uncontrollable factors. The occurrence of uncontrollable factors such as pre-existing conditions has already transpired and is, by definition, irreversible. Moreover, most patients undergoing PD are homebound, which limits clinical interventions. Consequently, it is imperative to explore more adaptable and acceptable methods for implementing intervention programs to effectively address these challenges. In clinical practice, we recommend an integrated internet-based management approach that focuses on controlling the risk factors. This approach combines the following: (1) comprehensive patient education to improve disease-related knowledge, (2) enhanced follow-up protocols with increased visit frequency, (3) multimodal digital interventions (supplementing traditional telephone/WeChat follow-ups with interactive retraining modules), and (4) targeted self-management skill development, all of which are particularly crucial for outpatient PD management and prevention of rehospitalization.

Implications for future research

Given the paucity of the study types, RCTs should be increased in the future to establish rigorous and scientifically sound interventional studies that investigate the effectiveness of nursing intervention programs in preventing or reducing the incidence of infections. In this review, most of the patients’ daily care was provided by themselves or their family members, and irregularity of caregivers was a risk factor. Therefore, the role of caregivers should not be neglected, and future intervention studies should focus on the caregiver group to play a role in monitoring patients. As most patients are treated at home and health education relies on outpatient or online follow-ups, future studies can combine online and offline health knowledge sharing by organizing patient exchange meetings, expanding follow-up methods, increasing frequency, and enriching follow-up content. In addition, the literature included in this review was limited, and more studies are needed to explore the potential influencing factors.

Limitations for research

Although our study included preliminary screenings across multiple countries, the final analyses were primarily drawn from studies in China and Malaysia because of insufficient data. This limitation may affect the generalizability of our findings. Additionally, differences in ethnicity, nationality, and region among the study participants, coupled with small sample sizes in some studies, resulted in significant heterogeneity between studies. Future studies should incorporate data from more countries to enhance the breadth and representation of the results. This study included only Chinese and English literature and did not include literature from other languages, which may have had an impact on the results. The exclusion of studies reporting non-significant results may have introduced a potential bias toward overestimating the effect sizes, although sensitivity analyses suggested that this did not materially alter our overall conclusions. Therefore, using larger sample sizes and more rigorous study designs can help reduce heterogeneity and improve the accuracy of results. To address the potential bias introduced by excluding non-significant studies, future meta-analyses should consider including all the relevant studies, regardless of their statistical significance. This approach provides a more comprehensive perspective of the evidence.

Conclusion

The results of this study showed that a combination of DM, insecure catheter placement, dialysis duration > 2 years, BMI > 20 kg/m2, mechanical strain, lack of proper care, irregular caregiver, and a history of catheter pulling were risk factors for the development of catheter-related infections in patients undergoing PD. Owing to the small number of relevant studies, the literature included in this review was mostly case-control studies, and it is suggested that more high-quality original studies should be included in the future. Healthcare professionals can use this review as a reference, for the early identification of high-risk groups in the development of catheter-related infections, to develop directed and purposeful clinical interventions to reduce the rate of catheter-related infections.

Supplemental Information

Supplemental Information 1 PRISMA checklist

Supplemental Information 2 Database Search Process

Supplemental Information 3 Data

Supplemental Information 4 List of excluded studies with rationale

Additional Information and Declarations

Competing Interests

Author Contributions

Data Availability

The authors declare there are no competing interests.

Juan Sun conceived and designed the experiments, analyzed the data, prepared figures and/or tables, and approved the final draft.

Miaomiao Zhao conceived and designed the experiments, analyzed the data, prepared figures and/or tables, and approved the final draft.

Yifang Zhang performed the experiments, prepared figures and/or tables, and approved the final draft.

Wenyu Zhang analyzed the data, authored or reviewed drafts of the article, and approved the final draft.

Huimin Zhang analyzed the data, authored or reviewed drafts of the article, and approved the final draft.

Qingzhu Bao conceived and designed the experiments, authored or reviewed drafts of the article, and approved the final draft.

The following information was supplied regarding data availability:

This is a systematic review/meta-analysis.

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
