# Peer review of "Factors influencing catheter-related infections in peritoneal dialysis patients: a meta-analysis"

_PeerJ, doi:10.7717/peerj.20063_

## Round 0.1 · original submission · Major Revisions

**Language Note:** The review process has identified that the English language must be improved. PeerJ can provide language editing services - please contact us at [email protected] for pricing (be sure to provide your manuscript number and title). Alternatively, you should make your own arrangements to improve the language quality and provide details in your response letter. – PeerJ Staff

Reviewer 1 ·

Basic reporting

The manuscript follows the usual structure for a meta-analysis, with a clear abstract, introduction, methods, background results and discussion. The figures and tables are relevant, clearly labelled, and help to support the findings. It is good to see that the authors have included PRISMA.checklist and that the review was pre-registered on Prospero. Thank you for sharing the raw data.

Areas for improvement: English language should be improved for clarity, especially when you start discussing the relevance for clinical practice. I suggest you have a colleague who is proficient in English and familiar with the subject matter review your manuscript, or contact a professional editing service.

Experimental design

I found the research gap to be well articulated: lack of evidence on risk factors for catheter-related infections in PD patients. The inclusion and exclusion criteria are well described, and the screening process involving two reviewers, with arbitration for disagreements is appropriate and reassuring.
Areas for improvement:
1. The inclusion of only case-control and cohort studies is appropriate; however, the decision to exclude RCTs should be justified.
2. The exclusion of studies showing non-significant results (line 124) could introduce bias. This requires discussion
3. The manuscript states (lines 122–123) that three studies were excluded due to an inability to extract data. Please specify what type of data was missing and whether any attempts were made to contact the original study authors. All exclusions should be clearly justified.
4. The limitation of including only English and Chinese language studies should be acknowledged earlier. This should also be justified.

Validity of the findings

Appropriate meta-analytic techniques were used: random/fixed-effects based on heterogeneity, sensitivity analysis, and Egger’s test. Heterogeneity is acknowledged and discussed, sensitivity analysis is thorough.
Areas for improvement:
1. The claim of “relatively comprehensive inclusion of influencing factors” is overreaching given the limitations acknowledged in the discussion (few foreign studies, limited diversity). Lines 231-232
2. Line 135, Egger is a proper name should be upper case, correct all appearances
3. Line 136, Authors say “in the Figure”. Please say which figure.
4. Lines 224-228, Divide paragraph in smaller sentences as it is too difficult to follow. It starts with “A total of 13 studies..” The same with next paragraph starting with “The methodological quality” (line 228). Please improve clarity and English.
5. Not sure why lines 235-237 are in bold.
6. Authors mention diabetes mellitus, but do not distinguish between type 1 and type 2, are they assumed to be the same? They could have different impacts in PD. Could you explain?
7. The paragraph beginning at line 348 should be revised for clarity, as the current wording is difficult to follow. Consider breaking it into shorter, more direct sentences. For example, the phrase “influential factors summarised have uncontrollable factors that have already occurred” is unclear; rephrase or explain what is meant.
8. Similarly, the paragraph beginning at line 351, starting with “Therefore,” is unclear and difficult to follow. consider rewriting it for improved clarity.

Additional comments

I would appreciate a more detailed discussion of the study’s limitations, including how these limitations may have influenced the findings. If possible, please elaborate on what you could have been done to address them.
The PRISMA checklist is mostly complete. However, some important elements are missing or insufficiently addressed, including item 13e (methods used to explore potential causes of heterogeneity among study results), item 22 (certainty of evidence), as well as items 20a and 27. Could you please justify why these items were not discussed or reported?

Reviewer 2 ·

Basic reporting

Line 49 spelling
Line 55 paragraph somewhat vague there have been studies examining risk factors and these should be described in more detail, in particular catheter care, prophylaxis used at exit sites – see ISPD infection guidelines with up to date evidence
ISPD peritonitis guideline recommendations: 2022 update on prevention and treatment - Philip Kam-Tao Li, Kai Ming Chow, Yeoungjee Cho, Stanley Fan, Ana E Figueiredo, Tess Harris, Talerngsak Kanjanabuch, Yong-Lim Kim, Magdalena Madero, Jolanta Malyszko, Rajnish Mehrotra, Ikechi G Okpechi, Jeff Perl, Beth Piraino, Naomi Runnegar, Isaac Teitelbaum, Jennifer Ka-Wah Wong, Xueqing Yu, David W Johnson, 2022
ISPD Catheter-related Infection Recommendations: 2023 Update

Experimental design

This is a very specific metanalysis of studies only from China except one form Malaysia no real rationale for only including these countries which limits the review, also why not include RCT'S need to give rationale
Line 62 Survey methodology such this be titled review methods- its not a survey

Validity of the findings

Irregular care givers not sure of this term for the point raised -is this not just about immobilisation and how is different to catheter fixation ?
Line 274 repeated information
Line 275 should be long term not longtime
Line 292 most retraining and monitoring of exit sites is done by nurses perhaps just change form medical personnel to health care staff the same for other references of medical personnel should be health care staff
Line 309 again refer to ISPD guidelines of recommendations for immobilisation of exit site

Additional comments

In general as all the evidence reviewed is from China albeit one study from Malaysia it should be reflected in the title

---

## Round 0.2 · Minor Revisions

Please carefully revise the grammar and phrases of the manuscript (maybe with a professional agency). In addition, try to resolve the reviewer's remaining concern.

**Language Note:** The Academic Editor has identified that the English language must be improved. PeerJ can provide language editing services - please contact us at [email protected] for pricing (be sure to provide your manuscript number and title). Alternatively, you should make your own arrangements to improve the language quality and provide details in your response letter. – PeerJ Staff

Reviewer 1 ·

Basic reporting

-

Experimental design

-

Validity of the findings

-

Additional comments

All comments have been addressed; thank you for taking the time to revise the manuscript.

The English has been improved, and the text reads well. I’m fully satisfied with the responses and revisions.

Reviewer 2 ·

Basic reporting

Thank you for taking the time to address reviewers' comments. The paper has improved; there are just a few minor grammatical errors to address.

Experimental design

-

Validity of the findings

-

Additional comments

Just re look at the paragraph on irregular caregivers, particularly this sentence

However, relatively few studies have been conducted on the factors influencing caregiver immobility.

What is this referring to? The caregiver's mobility status or immobility relating to the PD exit site is not clear.

I would suggest another read-through by an English-speaking editor to help.

---

## Round 0.3 · accepted · Accept

The manuscript is now much more clear and easier to be understood. No further comments.